# Effects of Nicotine on the Thermodynamics and Phase Coexistence of Pulmonary Surfactant Model Membranes

**DOI:** 10.3390/membranes14120267

**Published:** 2024-12-11

**Authors:** Fadi S. S. Magalhães, Ernanni D. Vieira, Mariana R. B. Batista, Antonio J. Costa-Filho, Luis G. M. Basso

**Affiliations:** 1Laboratório de Ciências Físicas, Centro de Ciência e Tecnologia, Universidade Estadual do Norte Fluminense Darcy Ribeiro, Avenida Alberto Lamego, 2000, Campos dos Goytacazes 28013-602, RJ, Brazil; fadi.simon.76@gmail.com; 2Laboratório de Física Biológica, Instituto de Física, Universidade Federal de Goiás, Avenida Esperança s/n, Campus Samambaia, Goiânia 74690-900, GO, Brazil; ernanni@ufg.br; 3School of Life Sciences, Gibbet Hill Campus, University of Warwick, Coventry CV4 7Al, UK; mariana.bunoro-batista@warwick.ac.uk; 4Laboratório de Biofísica Molecular, Departamento de Física, Faculdade de Filosofia, Ciências e Letras de Ribeirão Preto, Universidade de São Paulo, Ribeirão Preto 14040-901, SP, Brazil

**Keywords:** nicotine, pulmonary surfactant, phase coexistence, ESR, DSC, van’t Hoff, drug-membrane interactions, MD simulations

## Abstract

Phase separation is essential for membrane function, and alterations in phase coexistence by membrane-interacting molecules, such as nicotine, can impair membrane stability. With the increasing use of e-cigarettes, concerns have arisen about the impact of nicotine on pulmonary surfactants. Here, we used differential scanning calorimetry (DSC), molecular dynamics (MD) simulations, and electron spin resonance (ESR) to examine nicotine’s effect on the phase coexistence of two surfactant models: pure DPPC and a DPPC/POPC/POPG mixture. Our DSC analysis revealed that nicotine interacts with both membranes, increasing enthalpy and entropy change during the phase transition. ESR revealed that nicotine affects membrane fluidity and packing of DPPC more effectively than the ternary mixture, especially near the surface. MD simulations showed that neutral nicotine resides in the mid-plane, while protonated nicotine remains near the surface. Nicotine binding to the membranes is dynamic, switching between bound and unbound states. Analysis via ESR/van’t Hoff method revealed changes in the thermodynamics of phase coexistence, yielding distinct non-linear behavior. Nicotine altered the temperature dependence of the free energy, modifying the thermodynamic driving forces and the balance of non-covalent lipid interactions. These findings provide new insights into how nicotine influences pulmonary surfactant model membranes, with potential implications for surfactant function.

## 1. Introduction

Biological membranes consist of complex mixtures of proteins, carbohydrates, and lipids with different hydrocarbon chains, polar groups, backbone structures, chemical linkages, etc. [1]. Because of their complexity, studying biological membranes remains a major challenge from both experimental and theoretical perspectives. Therefore, a common strategy has been to investigate simple model membranes comprising only one lipid or mixtures of selected lipids to understand the membrane heterogeneity and intermolecular interactions puzzle.

Differential scanning calorimetry (DSC) is a powerful technique for studying the thermotropic phase behavior and phase diagrams of model and biological membranes, providing direct information on the thermodynamics of the systems [2,3,4]. DSC has also been used to investigate the effects of different additives in lipid thermal transitions, helping to elucidate the mechanism of action of drugs [5,6], peptides [7,8], lipid-binding proteins [9], peripheral membrane proteins [10], etc. However, the measure of heat flow only allows for a partial description of the thermotropic processes at a molecular level. Therefore, specific structural alterations related to inter- and/or intramolecular interactions should be addressed using other techniques.

Molecular dynamics (MD) simulations are a powerful tool to unravel the intricacies of drug-membrane interactions, shedding light on essential molecular-level details that are challenging to access experimentally [11,12]. By simulating the dynamic behavior of drug molecules in a water/lipid bilayer environment, MD can provide crucial insights into their conformational preferences, local ordering within the membrane, and the depth of penetration into the lipid bilayer [13,14]. Through MD, one can study concentration and charge dependence on the dynamic process of drug-membrane interactions as well as the stability and orientation of drugs within the membrane [14]. This wealth of information enables a deeper understanding of how drugs interact with cell membranes, influencing their efficacy, pharmacokinetics, and potential adverse effects.

Electron spin resonance (ESR) relies on introducing paramagnetic probes, such as stable nitroxide radicals, into otherwise diamagnetic systems. These probes are covalently attached to or associated with the molecule of interest, enabling the study of the molecular structure, dynamics, and interactions in intrinsically diamagnetic biological systems [15]. Spin-labeled phospholipids have been widely utilized to monitor changes in membrane fluidity, packing of the hydrocarbon chains, local polarity surrounding the spin label, and phase transition temperatures, among other parameters, in model and biological membranes [16,17,18]. The sensitivity makes ESR one of the methods of choice when one intends to probe the thermotropic behavior of important microscopic parameters, such as the local ordering and mobility of phospholipids in lipid bilayer membranes. The ability of ESR to track down alterations in the membrane dynamical structure can be further improved when ESR spectral simulations are included in the analysis [19,20,21]. Non-linear least-squares (NLLS) simulations are a powerful tool in decomposing multicomponent ESR spectra from spin probes partitioned into different phases that coexist in model and biological membranes [22,23,24]. If the partition coefficient of the probe is known, the equilibrium constant K between the lipid states of a two-phase membrane is calculated by determining the fraction of the spin-labeled lipids in both states directly from NLLS spectral simulations. Therefore, van’t Hoff analysis can be performed upon recording changes in the equilibrium constant triggered by an external stimulus such as temperature. By plotting ln K versus reciprocal temperature (1/T), one can thus spectroscopically obtain thermodynamic quantities such as the enthalpy change $\left(\Delta H\right)$, the entropy change $\left(\Delta S\right)$, the free energy change $\left(\Delta G\right)$, and the heat capacity change $\left(\Delta C\right)$ of the process under investigation, without the need to directly measure heat flows. Therefore, ESR/NLLS, in combination with van’t Hoff analysis, emerges as an effective tool for indirectly extracting thermodynamical information from the coexistence of phases that display different ESR observables, such as motional and/or structural properties in model and biological membranes.

We have previously applied this ESR/NLLS approach to study the thermodynamics of the ripple phase of pure dipalmitoyl phosphatidylcholine (DPPC) membranes and the phase coexistence region of a negatively-charged ternary mixture comprising DPPC, palmitoyl–oleoyl phosphatidylcholine (POPC), and palmitoyl–oleoyl phosphatidylglycerol (POPG), which mimics the pulmonary surfactant lipid matrix [25]. Van’t Hoff analysis yielded non-linear curves, i.e., the dependence of ln K on 1/T was not linear over the temperature range corresponding to the phase coexistence region of the membranes, implying a temperature dependence of the thermodynamic parameters. Non-linear van’t Hoff plots have also been observed in several biological applications [26,27,28,29].

In the present work, we seek to extend the previous method to study how small molecules, such as drugs, alter the thermodynamics, or more specifically, the van’t Hoff behavior, of phase coexistence regions in model membranes. Our focus is nicotine, a natural product found in the nightshade family of plants, known for its potent parasympathomimetic effects as an agonist at most nicotinic acetylcholine receptors [30]. In recent years, especially in the U.S., there has been a marked rise in the popularity of electronic cigarettes, also known as e-cigarettes or vaping devices [31,32]. These devices heat a liquid solution containing nicotine, producing an aerosol for inhalation. This growing trend in e-cigarettes among smokers and non-smokers has raised concerns about the health implications of nicotine exposure.

Although the mechanisms of pulmonary nicotine absorption have been well studied [33], little is known about nicotine’s impact on the non-covalent interactions that stabilize the first barrier of protection of the pulmonary alveoli. To comprehensively understand nicotine’s implications for lung health, it is crucial to investigate its effect on pulmonary surfactant membranes at the molecular level [34]. Understanding the molecular mechanisms of this interaction will provide valuable insights into how nicotine compromises the integrity and functionality of pulmonary surfactants.

Pulmonary surfactant membranes within the lungs’ alveoli play a vital role in respiratory function. These membranes (1) reduce the surface tension at the air-liquid, preventing alveolar collapse; (2) contribute to pulmonary host defense; and (3) modulate immune responses [35,36,37,38]. The pulmonary surfactant matrix is predominantly composed of phospholipids, constituting about 80% of its weight, with DPPC accounting for approximately 50–70% of this fraction. Other major lipid components include unsaturated zwitterionic POPC and anionic POPG [39]. Given the biological significance of these components, we have based our lipid matrices on our previous work, selecting pure DPPC and a negatively charged ternary mixture of DPPC/POPC/POPG (4/3/1 molar ratio) as representative models of pulmonary surfactant [25]. In such simplified model membranes, interactions of external molecules have provided valuable insights into changes in the membrane’s physicochemical properties these interactions can induce [40,41,42].

In this study, we employed DSC, molecular dynamics simulations, and ESR spin-label spectroscopy coupled with NLLS spectral simulations to investigate the effects of nicotine on the structural dynamics and the thermotropic behavior of the two-phase coexistence region in pulmonary surfactant model membranes. Our findings indicate that nicotine interacts weakly yet dynamically with the model membranes, with its specific location within the membrane dependent on its protonated state. Further analysis revealed that nicotine modifies the van’t Hoff behavior of phase coexistence, thereby altering the thermodynamic balance of non-covalent lipid interactions within the membrane, potentially compromising the integrity and functionality of pulmonary surfactants.

## 2. Materials and Methods

### 2.1. Reagents

The phospholipids 1,2-dipalmitoyl-*sn*-glycero-3-phosphocholine (DPPC), 1-palmitoyl-2-oleoyl-*sn*-glycero-3-phosphocholine (POPC), 1-palmitoyl-2-oleoyl-*sn*-glycero-3-phospho-(1′-rac-glycerol) (POPG), and the spin labels 1,2-dipalmitoyl-*sn*-glycero-3-phospho(tempo)choline (DPPTC) and 1-palmitoyl-2-stearoyl(n-doxyl)-*sn*-glycero-3-phosphocholine (n-PCSL, where n = 5 and 16) were purchased from Avanti Polar Lipids, Inc. (Alabaster, AL, USA). The (-)-1-methyl-2-(3-pyridyl)pyrrolidine (nicotine) and the spin labels 16-doxyl-stearic acid (16-SASL), and 16-doxyl-stearic acid methyl ester (16-MESL) were purchased from Sigma-Aldrich Brasil Ltd. (São Paulo, Brazil). All reagents were used without further purification.

### 2.2. Sample Preparation

Multilamellar vesicles of DPPC and DPPC/POPC/POPG (4/3/1 molar ratio) for DSC and ESR were prepared as described elsewhere [25]. Briefly, a measured volume of the nicotine stock solution in ethanol was added to the lipids in chloroform at different concentrations: 2, 4, 10, 20, and 50 mol%, corresponding to lipid/nicotine molar ratios of 50/1, 25/1, 10/1, 5/1, or 2/1. The organic solvent mixture was vortexed thoroughly and evaporated under an N2 flow, and the resulting lipid film was placed under a vacuum for 12 h. Pure lipid and lipid/nicotine samples were hydrated with a buffer containing 10 mM acetate, borate, phosphate, at pH values of 5.0, 7.4, or 11.0, containing 150 mM of NaCl, sonicated for 10 min at 45 °C in a bath-type sonicator, and freeze-thaw cycled ten times prior to the experiments.

### 2.3. Differential Scanning Calorimetry (DSC) Experiments

DSC measurements were carried out in a VP-DSC MicroCal microcalorimeter (Microcal, Northampton, MA, USA) using a heating rate of 18.1 °C/h for DPPC and 33.3 °C/h for DPPC/POPC/POPG. After 10 min of sample equilibration at the starting temperature, the thermograms were recorded from 20 to 55 °C for DPPC and from 6 to 60 °C for the ternary mixture until reversibility of all transitions and reproducibility of the thermograms were achieved (usually 3 to 4 heating scans). A faster scan rate was used for the ternary mixture to increase the signal-to-noise ratio due to the broadening of the thermograms and their lower heat capacity values compared to pure DPPC [4]. The concentration of DPPC was 2 mg/mL, and that of DPPC/POPC/POPG was 5 mg/mL. Data analysis was performed using Microcal Origin 2022 software. Briefly, the thermograms of the lipid systems were subtracted from the buffer baseline and normalized by the lipid concentration. The melting temperature $\left(T_{m}\right)$ was taken as the temperature corresponding to the maximum of the heat capacity $\left(C_{p}\right)$ curves. The calorimetric enthalpy change, $\Delta H_{cal}$ was calculated from the area under the $C_{p}$ curve, whereas the entropy change, ΔS, and the van’t Hoff enthalpy change, $\Delta H_{vH}$, were calculated as $\Delta S=\Delta H_{cal}/T_{m}$ and $\Delta H_{vH}=4RT_{m}^{2}C_{p,max}/\Delta H_{cal}$, respectively.

### 2.4. Molecular Dynamics (MD) Simulations

Molecular dynamics simulations of the nicotine-membrane interaction were performed using the NAMD package [43]. Different protonation states for nicotine and different membrane compositions were used. The CHARMM-GUI membrane builder functionality [44] was used to set up four systems: (1) Neutral nicotine (NTN) + DPPC bilayer; (2) protonated nicotine (NTH) + DPPC; (3) neutral nicotine + lung surfactant bilayer; (4) Protonated nicotine + lung surfactant membrane. The lung surfactant membrane was composed of DPPC/POPC/POPG at a 4/3/1 molar ratio. Different numbers of NTN or NTH molecules were initially placed above the membrane, and the systems were solvated with TIP3P waters and 0.15 M NaCl. MD simulations were performed using the CHARMM36 force field [45] with a timestep of 2 fs. The input files for minimization and equilibration provided by CHARMM-GUI were used. The system was first minimized, followed by a series of NVT and NPT equilibration steps consisting of gradual removal of the restraints from lipid atoms for a total time of 2 ns. A total of 80 ns of unrestrained atomistic MD simulations for each configuration of the molecular system were performed. The long-range electrostatic interactions were computed with the Particle Mesh Ewald (PME) method [46]. A cutoff of 12 Å was used for van der Walls interactions. The simulations were performed in the isothermal-isobaric ensemble at 1 atm, and two different temperatures, 310.15 K and 318.15 K. Temperature was controlled using Langevin dynamics [47] with a damping coefficient of 1.0 ps^−1^. The Nose-Hoover algorithm [48] was used for pressure control, with a piston oscillation period of 200 fs and a decay rate of 100 fs. Covalent bonds involving hydrogen atoms in the protein and lipids were constrained to their equilibrium distances using the SHAKE algorithm, while the SETTLE [49] algorithm was used for water. The simulations were analyzed using MDAnalysis [50], Loos library [51], and in-house protocols. All images were generated using Pymol [52]. The description of the systems and simulations performed are summarized in Appendix A.

### 2.5. Electron Spin Resonance (ESR) Measurements

ESR experiments were carried out in a Varian E109 spectrometer operating at X-band (9.4 GHz) using the following acquisition conditions: microwave power, 10 mW; field modulation frequency, 100 kHz; and field modulation amplitude, 0.2–0.5 G depending on the temperature range. The temperature was controlled using an E257-X Varian temperature control unit coupled to the spectrometer (uncertainty about 0.2 °C). The samples were transferred to glass capillaries (1.5 mm I.D.) and centrifuged at 10,000 rpm for 10 min. The capillaries containing the lipid pellets were then bathed in mineral oil (for thermal stability) inside an ESR quartz tube and placed in the center of the resonant cavity. Prior to measuring the ESR spectra, the sample was thermalized at each temperature for about three minutes. Typically, samples contained 0.5 mg of DPPC or DPPC/POPC/POPG and 0.5 mol% of nitroxide-labeled lipids relative to the total phospholipids. ESR samples were prepared using the pH 7.4 buffer.

### 2.6. Non-Linear Least-Squares (NLLS) ESR Spectral Simulations

NLLS fittings of the ESR spectra were performed using the Multicomponent by Christian Altenbach. The software is written in LabView (National Instruments, Austin, TX, USA) and can be freely downloaded from the following site: http://www.biochemistry.ucla.edu/biochem/Faculty/Hubbell/ (accessed on 5 November 2024). The simulation of the ESR spectra involves a set of parameters, which include the gyromagnetic g-tensor ($g_{xx}$, $g_{yy}$, $g_{zz}$) and the magnetic hyperfine splittings ($A_{xx}$, $A_{yy}$, $A_{zz}$), the components of the rotational diffusion tensor ($R_{xx}$, $R_{yy}$, $R_{zz}$), and the ordering potential U(Ω), which is written as a series of spherical harmonic functions with coefficients $c_{20}$, $c_{22}$, $c_{40}$, $c_{42}$, and $c_{44}$ [19,20]. The order parameter S is, in turn, calculated as:(1)$$S=\left\langle \frac{3\cos^{2}\theta -1}{2}\right\rangle =\frac{\int d\Omega D_{00}^{2}\exp(-U/k_{B}T)}{\int d\Omega \exp\left(-U/k_{B}T\right)}$$
where $k_{B}$ is the Boltzmann constant, and T is the temperature. In multilamellar vesicles, the nitroxide-labeled lipid is considered microscopically ordered but macroscopically disordered, and, therefore, the so-called microscopic order macroscopic disorder (MOMD) model is used [19,20]. The rotational diffusion tensor (*R*) of the spin-labeled lipid is described in terms of the local director of the bilayer, that is, it is calculated around axes parallel ($R_{//}$) and perpendicular ($R_{⊥}$) to the symmetry axis of the lipid acyl chain and the ordering potential, which describes the orienting influence of anisotropic fluids, such as membranes. The 16-PCSL incorporated in lipid bilayers generally presents $R_{//}\gg R_{⊥}$ [53], and, following a standard procedure [19,21], we set $R_{//}=10R_{⊥}$.

## 3. Results

### 3.1. Nicotine Increased the Enthalpy Change for Acyl Chain Melting on the Pulmonary Surfactant Membrane Models

Figure 1A shows representative DSC thermograms illustrating the thermotropic phase behavior of fully hydrated multilamellar DPPC vesicles in the absence and presence of nicotine at pH 7.4. The temperature dependence of the excess heat capacity of pure DPPC displays two endothermic transitions, a low-enthalpic and broad pre-transition centered at $T_{p}$ = 34 °C, and a very intense and more cooperative main phase transition centered at $T_{m}$ = 41 °C. The low-temperature peak arises from the transition of the solid-ordered lamellar gel phase, $L_{\beta^{\prime }}$, to the ripple gel phase, $P_{\beta^{\prime }}$, whereas the high-temperature peak arises from the conversion of $P_{\beta^{\prime }}$ to the fluid, liquid crystalline $L_{\alpha }$ phase. In the $L_{\beta^{\prime }}$ phase, the lipids adopt an all-trans configuration and display a tilt angle of about 30° with respect to the membrane normal [54], whereas in the $L_{\alpha }$ phase, the lipid chains are mostly disordered and display many trans and gauche isomerizations in their C-C bonds. The $P_{\beta^{\prime }}$ phase is characterized by periodic undulations (ripples) on the membrane surface and is likely formed by periodic arrangements of both $L_{\beta^{\prime }}$ and $L_{\alpha }$ lipid domains [54]. The transition temperatures agree well with our previous works [7,25].

The addition of nicotine perturbs the heat capacity profile of DPPC in a concentration-dependent manner (Figure 1A,B). The thermograms exhibit a systematic shift towards lower temperatures, indicative of the drug’s preferential localization within the $L_{\alpha }$ phase of the membrane. Indeed, using a simple binding model, we determined the ratio of the binding constants of nicotine to the gel $\left(K_{G}\right)$ and fluid $\left(K_{F}\right)$ phases as $\frac{K_{F}}{K_{G}}$ = 1.7 ± 0.3 (Appendix A), indicating that nicotine binds more effectively to the fluid phase than the gel phase. This preference for the $L_{\alpha }$ phase aligns with findings reported for the antimalarial drug primaquine by Basso et al. [5], which showed a similar affinity trend. Conversely, including trans-parinaric acid in the lipid matrix caused an opposing effect, indicating a better partition into the gel phase [55]. Nicotine induces slight changes in the chain-melting temperature of the membranes ($\Delta T_{m}$ < 1 °C), even at the highest nicotine concentration (50 mol%). This result implies that nicotine has a limited impact on bilayer packing. Notably, the cooperativity of the transition, assessed by the inverse of the linewidth at half intensity ($\Delta T_{m,1/2}$), is virtually the same for all conditions (Table 1). The cooperative unit, calculated from the ratio of the van’t Hoff enthalpy change ($\Delta H_{vH}$) to $\Delta H_{cal}$, shows minimal variation in the presence of nicotine (Table 1). This parameter indicates the average number of molecules in a cluster that melts cooperatively as a unit [5].

Previous studies have demonstrated the remarkable sensitivity of phosphatidylcholines pre-transition to external molecules present at the membrane/water interface [5,56]. Intriguingly, the nicotine binding exerts minimal influence on both the $T_{p}$ and the shape of the pre-transition curve (Figure 1A, inset). This result suggests that neither the structure of the DPPC head group [57] nor the membrane hydration [58,59] is significantly affected by nicotine adsorption on the bilayer surface.

The most pronounced effects induced by nicotine were observed in the calorimetric enthalpy change of the entire transition, $\Delta H_{cal}$ (Figure 1B), and in the corresponding entropy change, ΔS, at the melting temperature $T_{m}$ (Table 1). Both $\Delta H_{cal}$ and ΔS increase in a dose-dependent manner, albeit not exceeding 14% of the values observed for the nicotine-free vesicles. These findings indicate that more heat is needed to melt the acyl chains of nicotine-bound DPPC vesicles, and more entropy is gained in this process.

To understand the impact of nicotine on the thermal stability of a more complex model of pulmonary surfactant membranes, we carried out DSC measurements of DPPC/POPC/POPG vesicles. Figure 1C illustrates the heat capacity profile of the ternary mixture without and with 10 mol% of nicotine at different pH values (5.0, 7.4, and 11.0). The pH was varied to probe the interaction of distinct nicotine protonation states. Nicotine exhibits two pKa values that vary with temperature [60]. The pKa values of the pyrrolidyl and pyridyl groups ionization at 37 °C are 7.65 and 2.77, respectively [60] (Appendix A). Hence, nicotine is in equilibrium between a monoprotonated cation, with the positive charge centered on the pyrrolidine nitrogen atom, and a neutral state at physiological pH. Under acidic conditions, the equilibrium shifts toward the protonated forms of nicotine, while a basic environment favors the neutral state (Appendix A). As a result, nicotine exists predominantly in its monoprotonated form at pH 5, whereas it is largely neutral at pH 11 and coexists in equilibrium between the monoprotonated (~58%) and neutral (42%) forms at pH 7.4.

The DSC thermograms for DPPC/POPC/POPG membranes displayed a significantly broader pattern of peaks compared to the sharp transitions observed for pure DPPC (Figure 1C). This broadening arises from the lipid components in the mixture presenting very distinct transition temperatures. The unsaturation in the carbon chains of POPC and POPG makes their transition temperatures drop to ca. −2 °C, in contrast to the 41 °C melting temperature of DPPC. This leads to a very heterogeneous arrangement of the lipids within the membrane, reflecting the broader DSC profiles. Interestingly, the thermograms of DPPC/POPC/POPG membranes resemble those of Curosurf^®^, a modified natural and cholesterol-free surfactant derived from porcine lungs, which contains lipids and about 2% of the hydrophobic proteins SP-B and SP-C [4,6]. The *T_m_* values observed for DPPC/POPC/POPG membranes (28.1–29.1 °C) are similar to those found for Curosurf^®^ (27–28 °C). The thermodynamic parameters of the phase transitions in our surfactant model membranes exhibited slight pH dependency (Table 2). These results highlight the influence of lipid composition and environmental conditions on the structural and dynamic properties of surfactant membranes.

The DSC results show that nicotine increases the enthalpy change of the phase transition of the ternary lipid mixture in a pH-dependent manner: the higher the pH, the larger the $\Delta H_{cal}$ (Figure 1C,D, Table 2). This result suggests that more heat is needed to induce the transition from an ordered phase to the fluid phase of the nicotine-bound DPPC/POPC/POPG membrane compared to the empty vesicles and that the neutral nicotine is more effective than the monoprotonated drug in affecting the membrane. Budesonide, on the contrary, a glucocorticoid with a broad anti-inflammatory effect, significantly affect the heat capacity profile of Curosurf^®^, reducing the enthalpy change of the transition [6]. Our results also show that the melting temperature of the vesicles and the cooperativity of the transition are only mildly affected by the drug, suggesting a limited impact on bilayer packing (Table 2). The $T_{m}$ values found for the ternary lipid mixture agree well with our previous work [25].

Although DSC accounts for a full thermodynamic description of the lipid phase transition, it does not provide the molecular details related to structural and dynamical changes taking place during the phase transition, which requires molecule-based methods to complement the DSC information.

### 3.2. Nicotine Slightly Changes the Structural Dynamics of the Surfactant Model Membranes

ESR has been widely used to examine how temperature and external agents affect structural and dynamic rearrangements in lipids of both model and biological membranes [23,40,61,62]. Using nitroxide-labeled phospholipids, ESR allows the calculation of the order parameter, which provides insights into membrane packing, and the rotational diffusion rates, which reveal changes in membrane fluidity. We used the spin-labeled phospholipids 5-PCSL and 16-PCSL, bearing the nitroxide moiety at carbons C5 and C16 on the sn-2 acyl chains, respectively, to probe changes in the structural dynamics of the lipids in regions near the membrane surface and close to the bilayer center.

Figure 2 shows ESR spectra for 5- and 16-PCSL in hydrated multilamellar vesicles of DPPC at pH 7.4 and selected temperatures. Notably, nicotine induces a more pronounced alteration in the lineshape of the 5-PCSL spectra compared to the 16-PCSL spectra, indicating that nicotine primarily affects the apolar region of the bilayer near the membrane surface. This is evidenced by a decrease in the maximum hyperfine splitting ($2A_{max}$) and a narrowing of the central line in the 5-PCSL spectra, suggesting reduced membrane packing and increased fluidity. These nicotine-induced effects are also phase-dependent, since the spectra remain virtually unchanged for temperatures higher than DPPC melting temperature ($T_{m}$ around 41 °C). Appendix A illustrates the temperature dependence of both $2A_{max}$ and the linewidth of the central line $\left(\Delta H_{0}\right)$ for the 5-PCSL spectra with and without nicotine. Two distinct transitions were observed in the $\Delta H_{0}$ vs. temperature plot: one around 41 °C, associated with $T_{m}$, and another near 27 °C, corresponding to the onset temperature ($T_{ons,pre}$) of the pretransition (inset of Figure 1A). Nicotine reduces both $T_{m}$ and $T_{ons,pre}$, highlighting its subtle but measurable impact on the thermal behavior of the DPPC bilayer.

Some ESR spectra in the ripple gel phase of DPPC display a lineshape indicative of two distinct spectral components (see arrows in Figure 3A). This two-component feature is also observed in spectra of other nitroxide-labeled lipids in DPPC, including the stearic acid derivatives 16-SASL and 16-MESL (Appendix A), suggesting the presence of phase polymorphism. This observation implies the coexistence of microdomains with differing structural organization and dynamics within the membrane. Additionally, ESR spectra of 16-PCSL in DPPC/POPC/POPG also exhibit overlapping spectral features, corresponding to spin labels in two distinct environments (Figure 3B). These results align with our previous studies [5,25]. These overlapped spectra are better analyzed by NLLS spectral simulations, which proved particularly effective in distinguishing the relative population of spin probes partitioned between coexisting domains or phases, each with unique membrane packing and fluidity properties [23,24]. Indeed, fitting the ESR spectra presented in Figure 3 with a single spectral component was insufficient; introducing a second spectral population to the model significantly improved agreement between experimental and theoretical curves.

Appendix A shows the experimental and best-fit ESR spectra of 16-PCSL in hydrated multilamellar vesicles of DPPC, simulated using one and two spectral components, with and without nicotine, at various temperatures and two concentrations: 4 and 10 mol%. At temperatures above 40 °C, the ESR spectra displayed three narrow lines characteristic of spin probes in the disordered, liquid crystalline $L_{\alpha }$ phase. Conversely, below a specific temperature, which varied between 16 and 27 °C depending on the sample, the spectra exhibit broader lines typical of slow-motion probes in the solid–ordered, $L_{\beta^{\prime }}$ phase of DPPC. Between 26.8 °C and 40.1 °C for pure DPPC, 23.2 °C and 40.6 °C for the DPPC/nicotine 25:1 molar ratio, and 16.8 °C and 38.8 °C for the DPPC/nicotine 10:1 molar ratio, the spectra display overlapping broad and narrow components. Modeling these spectra with two components in these intermediate temperature ranges substantially improves the fit (Appendix A), indicating phase coexistence within the membrane. This result aligns with prior findings that the ripple phase of fully hydrated phosphatidylcholine bilayers is structurally heterogeneous, exhibiting characteristics of both $L_{\beta^{\prime }}$ and $L_{\alpha }$ phases [63,64,65,66]. Interestingly, the emergence of the second spectral component occurs at a temperature lower than the pretransition onset temperature, observed in our DSC data (inset of Figure 1A). This result underscores the sensitivity of a molecule-based technique such as ESR in detecting local structural changes prior to the macroscopic endothermic events that calorimetric techniques can capture. Notably, higher nicotine concentrations correspond to a lower ESR-detected onset temperature and expand the temperature range of the phase coexistence region relative to pure DPPC.

The order parameter (*S*) and the rotational diffusion rate ($R_{⊥}$), derived from the NLLS simulations, provide key insights into bilayer structural organization and membrane fluidity, respectively. Figure 4 shows the temperature dependence of these parameters as obtained from the 5-PCSL and 16-PCSL spectra in DPPC, along with the 16-PCSL spectra in the ternary lipid mixture. In general, nicotine has a more pronounced effect on the order parameter and rotational diffusion of 5-PCSL (Figure 4A,B) than on 16-PCSL in DPPC (Figure 4C,D), particularly within the gel and ripple gel phases. Consistent with previous observations, nicotine reduces *S* and increases $R_{⊥}$ (Figure 4A,B), suggesting a disordering effect in the bilayer regions monitored by 5-PCSL. Furthermore, the ESR data obtained with 16-PCSL indicate that nicotine has minimal influence on the main phase transition temperature, corroborating our DSC findings. However, it does broaden the temperature range for the two-domain coexistence region in DPPC. Additionally, nicotine appears to have no significant effect on the ordering and mobility of 16-PCSL in both DPPC (Figure 4C,D) and DPPC/POPC/POPG mixture (Figure 4E,F), even at high concentrations (10 mol%). These findings suggest that nicotine does not impact the structural dynamics of the 16-PCSL, implying that membrane packing and fluidity at the bilayer center remain unaffected by the drug.

In line with these observations, ESR spectra of DPPTC, a spin-labeled phosphatidylcholine with a nitroxide group attached to the headgroup region, recorded in DPPC and DPPC/POPC/POPG, exhibited negligible changes in the presence of nicotine (Appendix A), suggesting that nicotine does not perturb the ordering and dynamics of the lipid headgroup. Therefore, our ESR results indicate that nicotine binds weakly to the membrane, inserting itself in the apolar region near the fifth carbon of the acyl chain. This positioning allows nicotine to act as a lipid spacer, increasing membrane disorder specifically within the bilayer’s apolar region near the surface.

### 3.3. Van’t Hoff Analysis

Although nicotine does not significantly affect membrane packing and fluidity at the bilayer center as indicated by 16-PCSL spectra, this spin probe is highly sensitive to changes in the population distribution across distinct membrane environments. Variations in spectral populations imply shifts in the equilibrium constant of lipid distribution across distinct domains, thereby influencing the thermodynamics of coexisting domains or phases. To gain further insights into the thermodynamic properties of nicotine/membrane interactions, we employed a theoretical approach previously established in our study [25] to analyze data within the two-domain coexistence region. By examining the population of each spectral component, we can extract thermodynamic information about the nicotine/lipid system. The data can be treated based on a van’t Hoff analysis by measuring the equilibrium constant in the phase coexistence region across a range of temperatures. Letting *K* represent the equilibrium constant between lipids in the $L_{\beta^{\prime }}$ and $L_{\alpha }$ domains of the DPPC ripple phase or the two-domain coexistence region in the ternary lipid mixture, and *K_P_* the partition coefficient of 16-PCSL across both domains, we can express *K*(*T*) as [25]
(2)$$K\left(T\right)=K_{P}\frac{P\left(L_{\alpha }\right)}{P\left(L_{\beta^{\prime }}\right)}=\exp\left(-\frac{\Delta G}{RT}\right)$$
where $P\left(L_{\alpha }\right)$ and $P\left(L_{\beta^{\prime }}\right)$ denote the populations of 16-PCSL in the $L_{\alpha }$ and $L_{\beta^{\prime }}$ phases, respectively, as determined by NLLS spectral simulations, *R* is the universal gas constant, and Δ*G* is the free energy required to transfer a lipid from the ordered (gel or gel-like) to the disordered (fluid or fluid-like) state. A plot of ln *K* versus 1/*T* (van’t Hoff plot) yields thermodynamic parameters associated with the phase coexistence region. The van’t Hoff behavior is considered classical if the ln *K* versus 1/*T* plot is linear and nonclassical otherwise [67,68]. Deviation from linearity, observed in several systems [26,27,28], suggests that Δ*G*, enthalpy (Δ*H*), entropy (Δ*S*), and heat capacity (Δ*C*) are temperature dependent [67,68].

In our previous work, we demonstrated that the relationship between ln *K* and 1/*T* within the phase coexistence region of DPPC and DPPC/POPC/POPG pulmonary surfactant membranes is non-linear and can be expressed as follows [25]:(3)$$\ln K=\ln\left(\frac{P\left(L_{\alpha }\right)}{P\left(L_{\beta^{\prime }}\right)}\right)=a+\frac{b}{T}+\frac{c}{T^{2}}+\frac{d}{T^{3}}$$

In this equation, we assume that 16-PCSL partitions equally between the ordered and disordered membrane phases/domains, with a temperature-independent partition coefficient ($K_{P}=1$), as previously reported [24,66,69,70]. As demonstrated by Vieira et al. [25], deviations from unity in *K_P_* do not alter the shape of the van’t Hoff curves but merely shift Δ*G* and Δ*S* values. The empirical coefficients *a*, *b*, *c*, and *d* are estimated by fitting Equation (2) to the experimental ln *K* versus 1/*T* data, and the thermal behavior of the thermodynamic parameters are determined as previously described [25].

Figure 5 illustrates the van’t Hoff plots for the 16-PCSL embedded in the studied membranes. The temperature dependence of ln *K* for the DPPC was fitted with a cubic function, while the ternary lipid mixture followed a parabolic fit, consistent with previous observations [25]. The best-fit coefficients for the ln *K* versus 1/*T* curves are provided in the Appendix A. Although nicotine does not alter the shape of the van’t Hoff plot, it significantly shifts the equilibrium constant, thereby affecting the thermodynamics of the coexisting membrane phases/domains.

Nicotine induces changes in the van’t Hoff plots at lower temperatures due to the broadening of the phase coexistence region, indicating distinct thermal behavior within the spectral populations. The nicotine binding to the membranes induces the appearance of a population of disordered lipids at the onset temperature, where ESR first detects phase coexistence. Upon heating, the proportion of the disordered lipids in the coexisting domains/phases of the surfactant model membranes varies non-linearly, resulting in the distinct patterns shown in Figure 5.

The alteration in the van’t Hoff plots for 16-PCSL in nicotine-bound membranes also leads to distinct thermotropic behaviors of Δ*G*, Δ*H*, Δ*S*, and Δ*C* associated to the equilibrium between the ordered and disordered phases, as shown in Figure 6 and Figure 7. In DPPC, nicotine makes the transition from ordered to disordered lipid states more thermodynamically favorable between 27.5 °C and 40.0 °C, as indicated by the lower Δ*G* in this range (Figure 6A). Conversely, for DPPC/POPC/POPG mixture, nicotine raises Δ*G*, making lipid transfer from an ordered to a disordered state more challenging (Figure 7A). As observed in Figure 6A, Δ*G* approaches zero as the temperature nears *T_m_* for DPPC. For the ternary lipid mixture, however, Δ*G* approaches zero at two separate temperatures, likely due to the distinct *T_m_* values of DPPC (*T_m_*~41 °C) and POPC/POPG (*T_m_*~−2 °C) in the mixture.

Interestingly, the saddle point of the Δ*G* curves for both nicotine-free and nicotine-containing DPPC membranes (Figure 6A) aligns with the pre-transition temperature, *T_P_*, of the membrane systems (inset of Figure 1A). For the ternary lipid mixture, the maximum of the Δ*G* curve is situated approximately midway through the temperature interval associated with phase coexistence (Figure 7A). Notably, the temperature *T_max_*, corresponding to the maximum of Δ*G*, coincides with saddle points observed in the heat capacity profiles of DPPC/POPC/POPG membranes as measured by DSC (Figure 1C). Both temperatures can be determined by differentiating Δ*G* with respect to *T* and setting it to zero. Given that ${\left(\frac{\partial \Delta G}{\partial T}\right)}_{P}=-\Delta S$, the entropy change for the isobaric transfer of 16-PCSL from the solid-ordered gel phase to the disordered fluid phase equals zero at *T_P_*. This result suggests that the entropy gain from trans-to-gauche isomerization and the formation of membrane undulations in the ripple phase is counterbalanced by interactions between the head group of fluid lipids and solvent molecules, which contribute a negative Δ*S*.

Nicotine also significantly modifies the enthalpy and entropy changes within the two-phase coexistence region. For DPPC, both Δ*H* and Δ*S* values are positive across the phase coexistence temperature range. While pure DPPC shows a parabolic dependence of Δ*H* and Δ*S* on temperature, the presence of nicotine alters this concavity, making both curves less pronounced (Figure 6B,C). Δ*C* varies linearly from negative values below *T_p_* to positive values at higher temperatures, with nicotine reducing the steepness of this slope (Figure 6D). The concavity of the Δ*H* and Δ*S* curves is then derived from the slope of the Δ*C* curve, since $\Delta C={\left(\frac{\partial \Delta H}{\partial T}\right)}_{P}=T{\left(\frac{\partial \Delta S}{\partial T}\right)}_{P}$. Overall, these findings indicate that nicotine substantially alters the thermodynamic driving forces within the DPPC phase coexistence region by affecting the heat capacity between the gel and fluid phases. In the pure DPPC system, the gel-to-fluid transition below *T_m_* is predominantly entropically driven (Δ*H* > 0 and Δ*S* > 0).

For the ternary lipid mixture, both Δ*H* and Δ*S* vary linearly with temperature, in the presence and absence of nicotine (Figure 7B,C). These parameters shift from negative to positive values across the phase coexistence region, and the temperature where Δ*H* and Δ*S* are zero corresponds to the maximum of the Δ*G* curve, *T_max_*. The slope of the Δ*H* curve provides Δ*C*, yielding an approximately constant, positive value for the heat capacity change (Figure 7D). Taken together, these results suggest that nicotine influences the thermodynamic driving forces in the phase coexistence region of the ternary membranes by modifying the heat capacity between ordered and disordered domains. The nearly constant, positive Δ*C* implies that lipid transfer between ordered and disordered states within coexisting domains is either entropically driven (Δ*H* > 0 and Δ*S* > 0) for *T* > *T_max_*, or enthalpically driven (Δ*H* < 0 and Δ*S* < 0), for *T* < *T_max_*. Despite differences in the driving forces for both pulmonary lung surfactant model membranes, entropy–enthalpy compensation holds in the nicotine-free and nicotine-containing DPPC and DPPC/POPC/POPG membranes (Appendix A), suggesting that drug binding and subsequent membrane perturbation contribute minimally to the free energy changes between the ordered and disordered states, consistent with findings on membrane-interacting proteins [65].

### 3.4. Molecular Dynamics Simulations Reveal the Drug Location and a Dynamic Binding

To gain further insight into the possible location of nicotine within the membranes, we conducted 80 ns of unrestrained, fully atomistic molecular dynamics (MD) simulations for each nicotine/membrane system. By simulating nicotine’s dynamic behavior within a water/lipid bilayer environment, MD can provide crucial insights into the nicotine-lipid interactions at atomic level [13,14]. Simulations of pure DPPC were performed at 37 °C and 45 °C, while the ternary DPPC/POPC/POPG mixture was simulated only at 37 °C. For each system, three nicotine concentrations were used to mimic the lipid-to-drug ratios used in the experiments. Both neutral (NTN) and monoprotonated (NTH) nicotine molecules were constructed and examined through MD simulations. A summary of the MD systems can be found in Appendix A. In total, eighteen systems were simulated, each for 80 ns, resulting in a cumulative simulation time of 1.44 μs. Force field parameters for both NTN and NTH are detailed in Appendix A.

Figure 8 shows the atom density profiles of nicotine, lipids, and water molecules across the entire MD trajectories for both lipid bilayers at selected nicotine concentrations. The density profiles reveal that both neutral and monoprotonated nicotine permeate the water and membrane phases. In most systems, nicotine density is more pronounced in the water phase than in the bilayer core, suggesting a dynamic binding equilibrium between nicotine’s bound and unbound states. Nonetheless, the density maximum for neutral nicotine is located around the middle of the monolayer in both DPPC and DPPC/POPC/POPG systems, whereas the monoprotonated nicotine density peak is positioned closer to the bilayer surface. Appendix A illustrates MD snapshots of selected systems indicating the configuration and location of both neutral and charged nicotine molecules within the membranes.

The electrostatic interaction of NTH with the phosphate groups in the polar head region likely drives this surface localization. To investigate this, we calculated the radial distribution functions (RDFs) between the pyrrolidyl nitrogen of nicotine and the phosphorus atom of the phosphate groups of the lipids. The results, shown in Figure 9, reveal distinct interaction patterns for the two nicotine states. For NTH, multiple resolved peaks are observed, corresponding to different “solvation” shells. The most prominent peak at ~3.5 Å, representing the first solvation shell, indicates a close interaction between the positively charged pyrrolidyl nitrogen of NTH and the negatively charged phosphate groups in the lipid head region. In contrast, the RDF for NTN exhibits a single, broader peak at ~5.2 Å, suggesting a more distal interaction between the neutral form of nicotine and the lipid phosphate groups. Moreover, the higher intensity of the RDF peak for NTH compared to NTN further supports the stronger association between NTH and the polar head region of the lipids. The distances and peak intensities derived from the RDF analysis highlight the critical role of electrostatic forces in mediating the localization of NTH at the lipid/water interface.

To evaluate whether NTN and NTH bind stably or dynamically to the membranes, we analyzed time-dependent contacts between nicotine, lipid molecules, water, and other nicotine molecules. Contacts were defined whenever atoms of two molecules (including hydrogen) were within 3 Å. Appendix A depict the temporal evolution of contact percentages for NTN and NTH in both DPPC and the ternary mixture. Results indicate that both neutral and monoprotonated nicotine exhibit dynamic membrane binding, switching between bound and unbound states. Additionally, neutral nicotine can interact with itself, forming clusters of NTN molecules.

## 4. Discussion

In this study, we extended our previous ESR/van’t Hoff analysis of lipid model membranes exhibiting $L_{\beta^{\prime }}$–$L_{\alpha }$ coexistence [25] to examine the interaction of a small drug-like molecule, nicotine, with a two-phase membrane. Our findings indicate that nicotine’s interaction with DPPC and DPPC/POPC/POPG bilayers alters the temperature dependence of the thermodynamic parameters (Δ*G*, Δ*H*, Δ*S*, and Δ*C*) in the phase coexistence region.

Nicotine, a monoprotonated cation at physiological and mildly acidic pH, carries a positive charge on the nitrogen atom in the pyrrolidine ring [60]. Its adsorption to DPPC membranes is restricted to the apolar region near the membrane-water interface, likely involving the pyridine ring insertion into the bilayer in a wedge-like orientation, as the pyridine and pyrrolidine rings are oriented nearly perpendicular to each other [71]. Nicotine does not notably affect the structure of the DPPC head group or the overall hydration of the bilayer, as indicated by the lack of significant effects on the lipid pre-transition. However, the positively charged pyrrolidine nitrogen may engage in electrostatic interactions with the negatively charged phosphate groups and ester groups within the lipid head group, potentially displacing water molecules from the membrane surface. While water displacement may occur, this effect could be counterbalanced by nicotine’s promotion of the fluid lipid population, which binds more water molecules than gel-state lipids [72]. Consequently, monoprotonated nicotine can modify the hydrogen-bonding network within the polar head group region and may stabilize defects at the membrane-water interface [73]. These interfacial interactions may contribute to the entropy and enthalpy changes observed between gel and fluid lipids in the presence of nicotine. Additionally, the acyl chain contribution to both Δ*H* and Δ*S* in the phase coexistence region is positive due to nicotine’s promotion of fluid lipids, increasing the gauche-to-trans conformer ratio relative to nicotine-free membranes.

Emerging evidence suggests that the unique composition of biological membranes promotes lateral segregation of their constituents into liquid-disordered and raft-like, liquid-ordered phases. The latter, enriched in proteins, cholesterol, and sphingolipids, plays a crucial role in various dynamic cellular processes [74]. Notably, the pulmonary surfactant matrix exhibits lateral phase separation up to physiological temperatures [75], with ordered/disordered phase coexistence critical for its biophysical function [76]. Interactions with external molecules such as drugs, amphiphiles, or proteins can disrupt surfactant membranes’ structure and fluidity, potentially disturbing the ordered-disordered domain equilibrium and leading to phase coexistence loss. For instance, the surfactant inhibitor C-reactive protein has been shown to bind to surfactant membranes, increasing membrane fluidity and abolishing phase coexistence, effectively inactivating lung surfactant [77]. Nicotine has been shown to alter surfactant mechanics by interacting with lipid head groups, increasing membrane compressibility and disturbing lipid packing, as reported in studies on models of surfactant monolayers [78]. These effects are particularly relevant under physiological conditions, where nicotine intercalates more deeply into the membrane, potentially impairing lipid-protein interactions crucial for surfactant functionality. Our findings further suggest that nicotine preferentially localizes within fluid lipid domains, reducing membrane packing of the phase coexistence region of DPPC and altering the thermodynamic equilibrium between ordered and disordered phases of surfactant membrane models. Nicotine’s disruption of lipid organization could impair surfactant proteins’ activity, potentially reducing surfactant functionality. Therefore, we propose that our thermodynamic ESR method offers a valuable approach to understanding the thermodynamics of complex biological membranes with lateral phase separation, providing a means to investigate how external agents influence the driving forces governing phase equilibrium.

## 5. Conclusions

We employed DSC, MD simulations, and ESR along with NLLS spectral simulations to study the interaction of nicotine with two pulmonary surfactant model membranes: pure DPPC and DPPC/POPC/POPG (4:3:1 molar ratio). Our findings indicate that nicotine binds weakly and dynamically to these membranes, fluctuating between bound and unbound states. While nicotine does not significantly alter the melting profile of the membrane, it increases the enthalpy and entropy associated with the acyl chain melting transition in both DPPC and DPPC/POPC/POPG systems. ESR spectral simulations further revealed that nicotine only marginally affects the structure and dynamics near the bilayer center in both ordered and disordered phases, likely due to its localization within the apolar region of the monolayer, close to the membrane surface. However, nicotine broadens the phase coexistence region and modifies the van’t Hoff behavior within the ordered-disordered coexistence region of both surfactant model membranes. Our ESR/NLLS/van’t Hoff analysis demonstrated that nicotine changes the thermodynamic driving forces and alters the balance of non-covalent lipid interactions in pulmonary surfactant membranes. In pure DPPC, the transition from ordered to disordered states is predominantly entropically driven, whereas in DPPC/POPC/POPG, it is driven by both entropy and enthalpy, depending on temperature. The thermal behavior of heat capacity in the phase coexistence region suggests that nicotine-lipid interactions alter the relative contributions of lipid conformational entropy, electrostatic interactions, and hydrogen bonding within the lipid head group region. We believe this thermodynamic ESR approach can be extended to study the interactions of membrane-active molecules with more complex models and biological membranes, particularly where the coexistence of different domains is detectable by ESR.

## Figures and Tables

**Figure 1 membranes-14-00267-f001:**
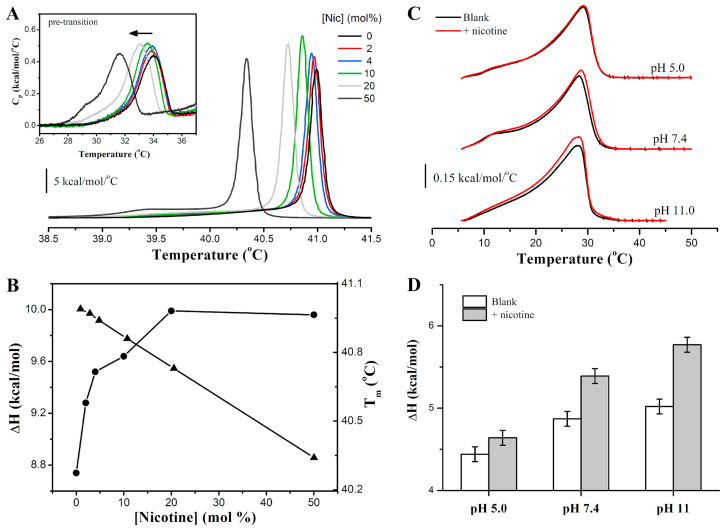
Thermotropic phase behavior of the surfactant model membranes as monitored by DSC. (**A**) Temperature dependence of the molar heat capacity of DPPC in the absence and presence of different concentrations of nicotine at pH 7.4. The inset illustrates the DPPC pretransition. (**B**) Effect of nicotine on the enthalpy change (ΔH, circles) and the melting temperature ($T_{m}$, triangles) of DPPC lipid vesicles. (**C**) Temperature dependence of the molar heat capacity of DPPC/POPC/POPG in the absence and presence of 10 mol% of nicotine at different pH values. (**D**) Impact of nicotine on the enthalpy change (ΔH) of the lipid mixture.

**Figure 2 membranes-14-00267-f002:**
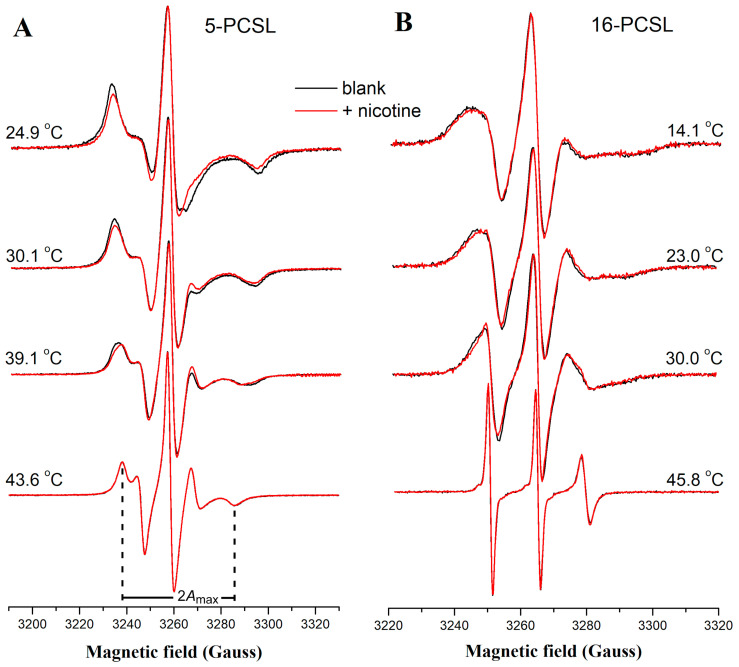
ESR spectral changes induced by nicotine. Representative ESR spectra of (**A**) 5-PCSL and (**B**) 16-PCSL in DPPC in the absence (black) and presence (red) of 10 mol% of nicotine at selected temperatures. The maximum hyperfine splitting, $2A_{max}$, is defined as the distance between the high-field minimum and the low-field maximum on the 5-PCSL spectrum. ESR samples were prepared using a pH 7.4 buffer.

**Figure 3 membranes-14-00267-f003:**
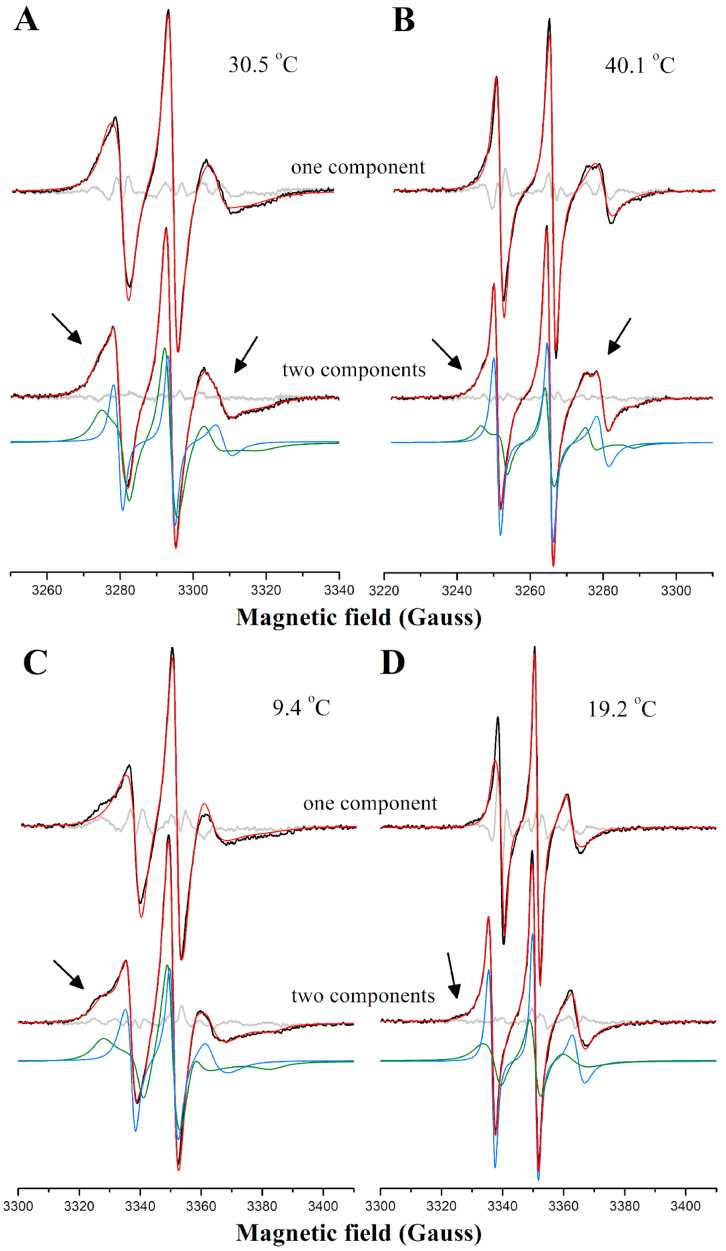
NLLS spectral simulations of ESR spectra displaying characteristic two-component features. Experimental (black) and best-fit (red) ESR spectra of 16-PCSL in (**A**,**B**) DPPC and (**C**,**D**) DPPC/POPC/POPG at selected temperatures and at pH 7.4. The fitting of the signals used one (**top**) or two (**bottom**) spectral components, the latter shown in green and blue lines. Gray lines represent the difference between the experimental and best-fit spectra. Arrows point to spectral features characteristic of a second spin population displaying different ordering or mobility.

**Figure 4 membranes-14-00267-f004:**
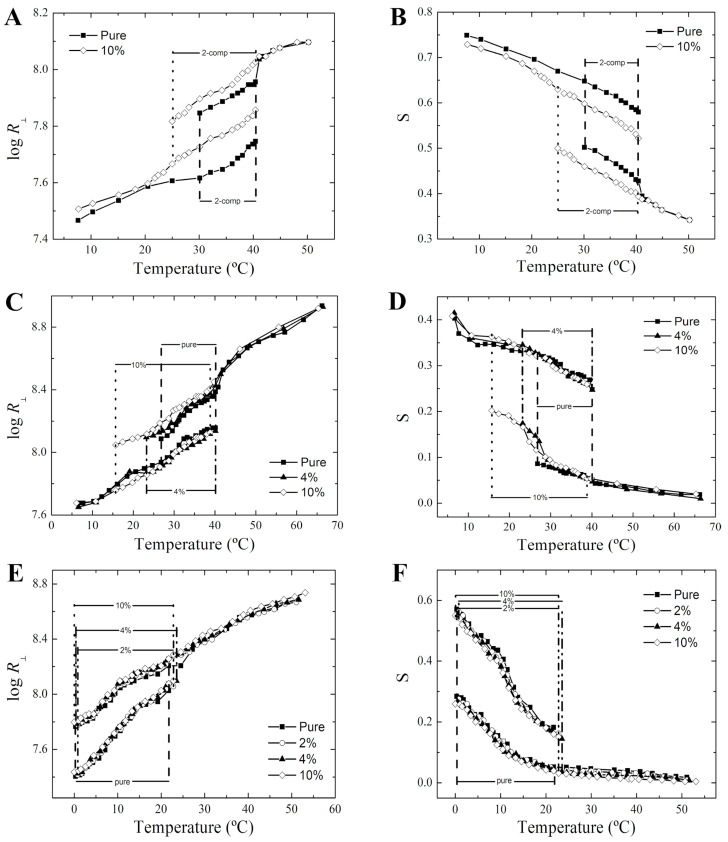
Thermotropic phase behavior of the membranes as monitored by ESR. Temperature dependence of the rotational diffusion rate $R_{⊥}$ (**A**,**C**,**E**) and the order parameter *S* (**B**,**D**,**F**) of 5-PCSL (**A**,**B**) and 16-PCSL (**C**–**F**) in DPPC (**C**,**D**) and DPPC/POPC/POPG (**E**,**F**) in the absence (full squares) and presence of nicotine at different concentrations: 2 mol% (empty circles), 4 mol% (full triangles), and 10 mol% (empty diamonds). Regions highlighted between dashed lines correspond to the phase coexistence. All experiments were performed using a pH 7.4 buffer.

**Figure 5 membranes-14-00267-f005:**
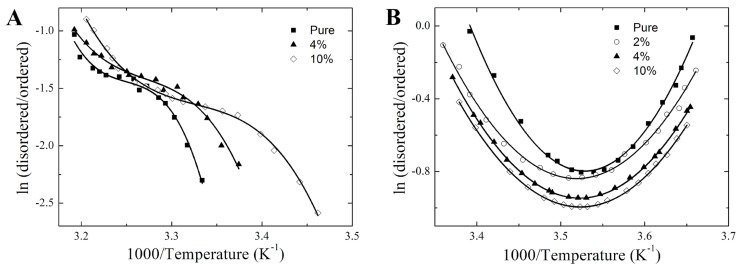
Nonlinear van’t Hoff behavior of pulmonary surfactant model membranes. Van’t Hoff plots illustrating the thermodynamic profiles of (**A**) pure and nicotine-enriched DPPC, and (**B**) pure and nicotine-embedded DPPC/POPC/POPG. The solid lines best fit the van’t Hoff plots, yielding cubic functions for DPPC and quadratic functions for the ternary model membranes. The molar percentage of nicotine relative to the amount of lipids is indicated. The adjusted-R² values obtained from the non-linear least-squares fitting ranged from 0.985 to 0.997 for DPPC curves and from 0.991 to 0.999 for the ternary mixture curves.

**Figure 6 membranes-14-00267-f006:**
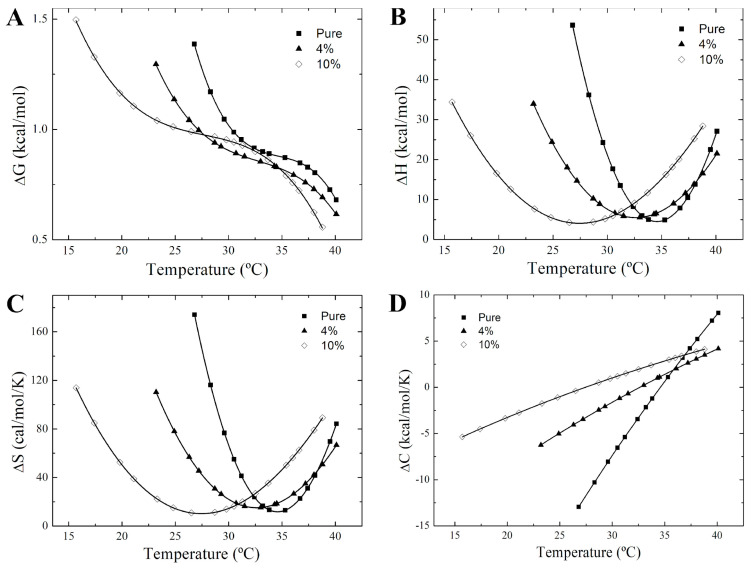
Thermodynamics of the phase coexistence region of DPPC and DPPC/nicotine multilamellar vesicles. Temperature-dependence of the changes in the (**A**) Gibbs free energy, Δ*G*, (**B**) enthalpy, Δ*H*, (**C**) entropy, Δ*S*, and (**D**) heat capacity, Δ*C* for pure DPPC (full squares) and nicotine-containing membranes at two concentrations: 4 mol% (full triangles), and 10 mol% (empty diamonds). The solid lines are guides for the eye.

**Figure 7 membranes-14-00267-f007:**
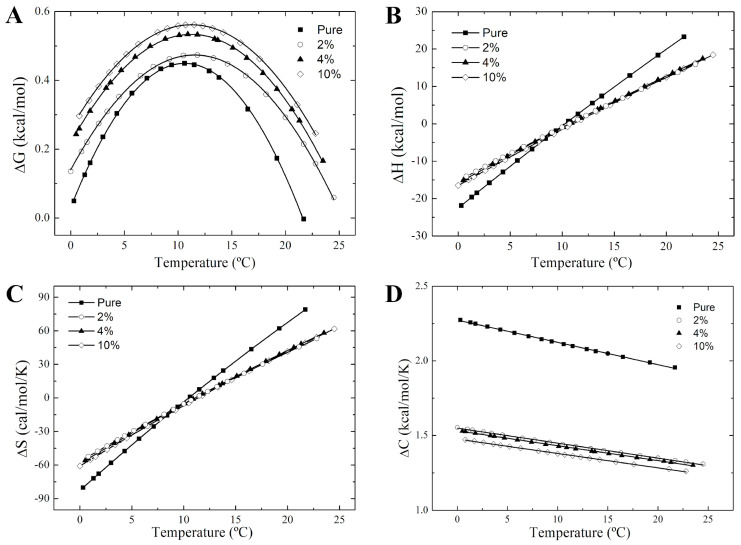
Thermodynamics of the phase coexistence region of nicotine-free and nicotine-containing DPPC/POPC/POPG multilamellar vesicles. Temperature-dependence of the changes in the (**A**) Gibbs free energy, Δ*G*, (**B**) enthalpy, Δ*H*, (**C**) entropy, Δ*S*, and (**D**) heat capacity, Δ*C* for pure DPPC/POPC/POPG (full squares) and nicotine-containing membranes at different concentrations: 2 mol% (empty circles), 4 mol% (full triangles), and 10 mol% (empty diamonds). The solid lines are guides for the eye.

**Figure 8 membranes-14-00267-f008:**
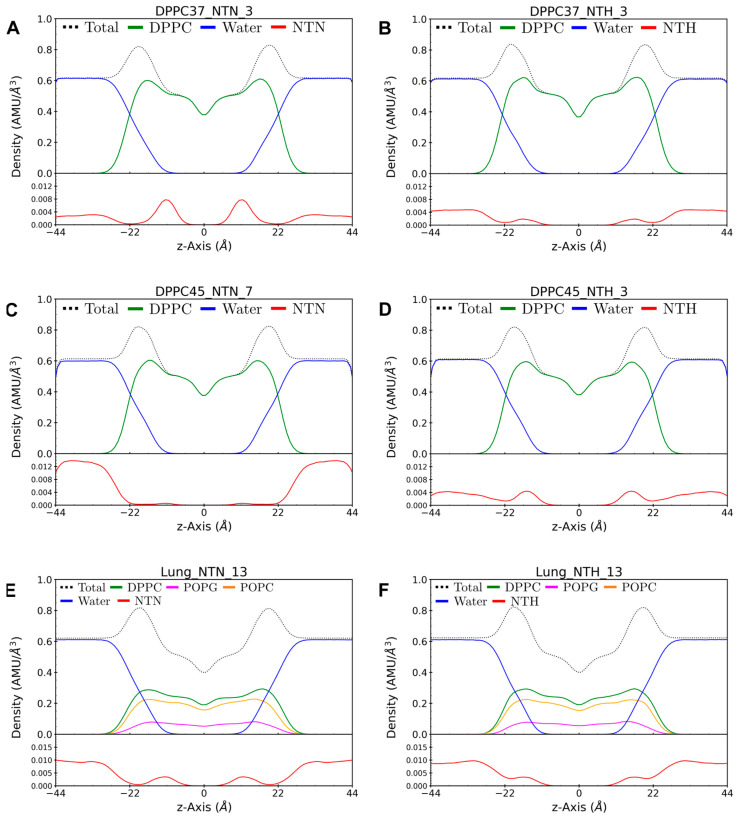
Spatial distribution of nicotine in membranes revealed by molecular dynamics simulations. Symmetrized atom density profiles with respect to the lipid bilayer normal for lipids, water, and both neutral (NTN, **left panels**) and monoprotonated (NTH, **right panels**) nicotine. Panels (**A**,**B**) depict the results for DPPC at 37 °C, panels (**C**,**D**) at 45 °C, while panels (**E**,**F**) represent the data for the lung surfactant model membrane (DPPC/POPC/POPG) at 37 °C. The numbers 3, 7, and 13 following the designation of each system on the top of the panels refer to the number of nicotine molecules in the system. To accommodate the varying scales, the upper graph in each panels displays the density profile for lipids and water, while the lower graph illustrates the nicotine distribution.

**Figure 9 membranes-14-00267-f009:**
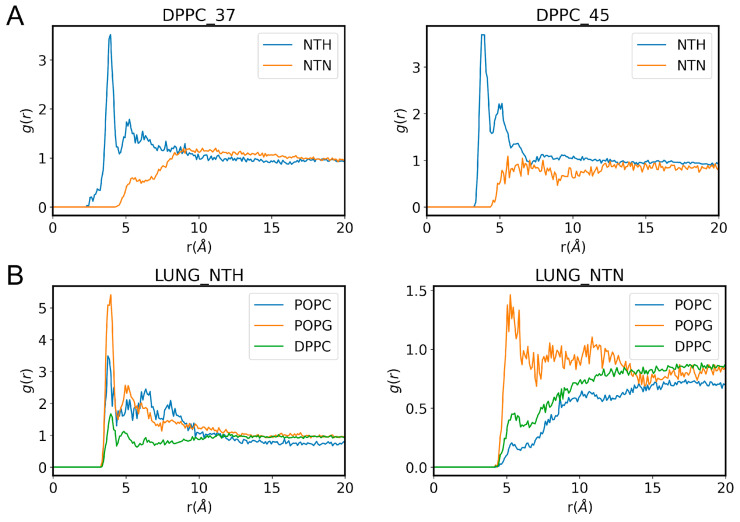
Radial distribution functions between the pyrrolidyl nitrogen of nicotine and the phosphorus atoms of lipid phosphate groups in (**A**) DPPC and (**B**) DPPC/POPC/POPG systems. Panel (**A**) shows the RDFs for NTN and NTH in DPPC bilayers at 37 °C (**left**) and 45 °C (**right**). Panel (**B**) illustrates the RDFs for NTH (**left**) and NTN (**right**) in the lung surfactant model membrane, highlighting the contributions from the phosphate groups of DPPC, POPC, and POPG phosphate groups.

**Table 1 membranes-14-00267-t001:** Calorimetric parameters obtained from the DSC thermogram of DPPC lipid vesicles in the absence and presence of nicotine at different concentrations (in mol%) and at pH 7.4. $\Delta H_{cal}$ represents the enthalpy change of the whole phase transition curve; $T_{m}$ is the main phase transition temperature; $T_{p}$, the pretransition temperature; $\Delta T_{m,1/2}$, the width at half height of the main phase transition peak; ΔS represents the entropy change of the transition at $T_{m}$; $\Delta H_{vH}$, the van’t Hoff enthalpy; and CUS, the cooperative unit size, given in number of molecules.

[Nic] (mol%)	Δ*H_cal_* (kcal/mol)	*T_m_* (°C)	Δ*T_m_*_,1/2_ (°C)	*T_p_* (°C)	Δ*S* (cal/mol/K)	Δ*H_vH_* (kcal/mol)	CUS
0	8.74	41.00	0.14	34.0	27.8	2930	335
2	9.28	40.97	0.13	34.0	29.5	2990	322
4	9.52	40.94	0.13	33.9	30.3	2980	312
10	9.64	40.86	0.12	33.6	30.7	3250	337
20	9.99	40.73	0.13	33.1	31.8	2990	299
50	9.96	40.34	0.13	31.6	31.8	2740	275

Estimated errors: Δ*H_cal_* (0.05 kcal/mol), *T_m_* (0.02 °C), Δ*T_m_*_,1/2_ (0.03 °C), *T_p_* (0.1 °C), Δ*S* (0.2 cal/mol/K), Δ*H_vH_* (20 kcal/mol).

**Table 2 membranes-14-00267-t002:** Calorimetric parameters obtained from the DSC thermograms of DPPC/POPC/POPG membranes without and with 10 mol% of nicotine.

Sample	Δ*H_cal_* (kcal/mol)	*T_m_* (°C)	Δ*T_m_*_,1/2_ (°C)
Blank, pH 5.0	4.44	29.1	7.7
+nicotine	4.64	29.2	8.0
Blank, pH 7.4	4.87	28.3	7.9
+nicotine	5.39	28.7	8.3
Blank, pH 11	5.02	28.1	8.6
+nicotine	5.77	28.2	8.7

Estimated errors: Δ*H_cal_* (0.09 kcal/mol), *T_m_* (0.1 °C), Δ*T_m_*_,1/2_ (0.1 °C).

## Data Availability

The raw data supporting the conclusions of this article will be made available by the authors on request.

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
