# Peer review of "Effects of Nicotine on the Thermodynamics and Phase Coexistence of Pulmonary Surfactant Model Membranes"

_membranes, 2024, doi:10.3390/membranes14120267_

Round 1

Reviewer 1 Report

Comments and Suggestions for Authors

The paper presents important results of the effects of nicotine on the thermodynamics and phase coexistence of pulmonary surfactant model membranes. The Authors show a breadth of different experimental and computational techniques including differential scanning calorimetry (DSC), electron spin resonance (ESR) and molecular dynamics (MD) simulations, to examine the effects of nicotine on two surfactant model systems: one consisting of  pure DPPC and the other with a ternary mixture of DPPC/POPC/POPG lipids. The results presented in this paper are, in the reviewer’s opinion, significant and worthy of publication. 

The manuscript would benefit, however, with some improvements to readability. 

 ·      Figure 1 – A better quality figure would be beneficial.

 ·      Line 281 – I clearly missed it. How did the Authors arrive at the conclusion that: “[a] more acidic environment increases the protonated form, whereas a more basic one shifts the equilibrium toward the neutral state”?

 ·      Line 320 the terms 5-PCSL and 16-PCSL are not defined.

 ·      Line 324: 2Amax – would make discussion clearer if indicated in Figure 2.

 ·      Line 380: the order parameter (S) and rotational diffusion rate (𝑅) are not defined.

 ·      Lines 407-414 (last paragraph on page 10) – perhaps the Authors could show the data in the supplement, even if the data “exhibited negligible changes in the presence of nicotine”.

In several places, the Authors engage in speculations without presenting evidence. They are not needed as the data shown in the manuscript warrants publication, for example:

·      Line 367 “[t]his observation may be attributed to conformational changes in 16-PCSL likely driven by microscopic rearrangements in the surrounding lipid environment”

·      Line 481: “[…] likely due to the distinct Tm values of DPPC”

·      Line 510: “[t]his effect likely arises from differences in heat capacity changes between gel and fluid domains”

·      Line 573: “[t]he electrostatic interaction of NTH with the phosphate groups in the polar head region likely drives this surface localization”.

·      Line 601: “[t]hese alterations are likely driven by changes in hydrogen-bonding properties and lipid conformational entropy”

·      Line 616: “[t]hese interfacial interactions likely contribute to the entropy and enthalpy changes observed between gel and fluid lipids in the presence of nicotine”.

Author Response

Please, see the attached file

Reviewer 2 Report

Comments and Suggestions for Authors

The manuscript by Magalhaes et al. provides a detailed thermodynamic study of the effect of the nicotine molecule on two lung surfactant model systems. Three methods of study were involved to gain a complex view of the investigated systems. In general, the study appears to have been conducted carefully and the experimental data obtained are reliable. Particularly remarkable is the comprehensive ESR data analysis.

Main comments:

1. Incomplete description of the differences between the two lipid systems used: the ternary mixture possesses, contrary to the pure DPPC system, a negative surface charge. This should be mentioned in both the introduction and the discussion of the results obtained, as it could also account for the differences observed during the interaction with the charged form of nicotine. At the same time, a precise description of the DSC scan obtained for the ternary mixture is lacking. It is necessary to explain the significant broadening of the heat capacity profile compared to pure DPPC.

2. Insufficient discussion/comparison of the results obtained with other studies focused on the effect of nicotine on biophysical properties of lung surfactant membrane models. What are the implications for native lung surfactant with protein components?

Further comments/suggestions:

1. The articles listed in the Introduction applying DSC to study the effects of additives/drugs on lipid thermal transitions should be updated with recent DSC studies focusing on additive/drug effects on simple and complex lung surfactant models such as Curosurf.

2. What is the reason for the different heating rates used for the two systems in DSC measurements? How can the heating rate affect the results obtained? This should be stated in the manuscript.

3. How was Tm determined from DSC thermograms? Was a fitting procedure used? What is the error in the Tm determination? The latter may be crucial, as the observed changes in Tm are within 1°C. But, according to the thermograms in Fig. 1A inset, the changes in Tp are more significant. These values could also be included in Fig. 1B, for instance by plotting DTf on the y-axis.

4. Was there any specific reason for choosing 10 mol% nicotine to study the effect of pH on the nicotine-membrane interaction by DSC? I would assume to perform this study at 50 mol%, where the effect was most dominant at neutral pH. And was there any specific reason for choosing the 25:1 and 10:1 molar ratios to study the effect of the nicotine-membrane interaction by ESR? 

5. Information on the pH values of the samples prepared for ESR experiments is not given, please complete. How would pH affect the analysed thermodynamic parameters?

6. What is the partition coefficient Kp of nicotine? How does this value depend on the physical state of the membrane? Do these values indicate intercalation of the molecule into the membrane or rather surface interactions? Thus, are the obtained results consistent with the assumption?

7. Is there any phys-chem explanation for the difference in trends in Fig. 5 for the two studied systems? Is there any phys-chem justification for the quadratic and cubic fits, respectively?

Technical notes:

1. Designation of the sample composition is inconsistent among the manuscript (both in the text and in the figures); the authors alternately use the molar ratio as mol:mol or mol/mol and mol%.  This should be unified.

2. In Fig. 4 plots B, D, F, the y-axes could be labelled “S” in accordance with description of the y-axes in plots A, C, E, because both “R” and “S” are explained in the Figure description.

3. Do the lines in Figs. 6D and 7B, C , D represent the best linear fits of the plotted data? The authors state that these thermodynamic parameters vary linearly, so the accuracy of the fit should be mentioned.

4. In Fig. 8 and Figures in the Supplementary material, the designation “1”, “3”, “7”, “13” as a number of nicotine molecules in the system is missing in the figure description.

Author Response

Plase, see the attached file.

Reviewer 3 Report

Comments and Suggestions for Authors

This manuscript provides data from a detailed study analyzing the interaction of nicotine with model phospholipid membranes intended to mimic pulmonary surfactant layers. The authors have used an appropriate combination of methodologies including DSC, EPR and computer simulation to describe that nicotine interacts in a more or less subtle way with phospholipid membranes, affecting the acyl chain region of phospholipids in segments close to the headgroups. This interaction is well described in terms of thermodynamic parameters.

The study seems to have been carried out well and carefully, and it is a natural continuation of other studies by a research group with large experience in the physico-chemical characterization of membranes and membrane/drug systems.

A few questions could be addressed to complete the relevance of the paper:

1.     To facilitate the readers to follow the rational and the discussion of some of the experiments, it could aid including in a figure the structure and charge of the nicotine molecule, as well as its potential configuration and location into the membranes, perhaps by adding some snapshots from the molecular simulation experiments.

2.     It is stated (line 63) “Electron spin resonance (ESR) of diamagnetic biological molecules…”. In this case, the authros have not studied diamagnetic biological molecules, but diamagnetic derivatives of biological molecules.

3.     “sn” in the nomenclature of the lipids used should be in ithalic according to IUPAQ.

4.     Why heating rate used in the DSC experiments has been different for DPPC and DPPC/POPC/POPG bilayers? Please clarify.

5.     Some comments regarding the potential effect of the inclusion of the spin probes into the membranes with respect to the phase coexistence are warrantees. Perhaps experiments with different proportions of the probes could have been aided to assess their potential perturbing effects.

6.     The most important question to address is that the whole study has been set with the goal to analyze the potential effect of nicotine to perturb the properties and activities of the endogenous pulmonary surfactant system, considering that nicotine is typically inhaled by customers of e-cigarettes (and of tobacco, one would say). However, after the detailed analysis of the thermodynamic effects of the nicotine/membrane interaction, no discussion has been devoted to suggest to what extent the effects seen could support deleterious effects on surfactant and on the respiratory system, considering the level of exposure to nicotine in smokers.

Author Response

Plase, see the attached file.
